# The chloroalkaloid (−)-acutumine is biosynthesized via a Fe(II)- and 2-oxoglutarate-dependent halogenase in Menispermaceae plants

Colin Y. Kim [1,2], Andrew J. Mitchell[1], Christopher M. Glinkerman[1], Fu-Shuang Li[1], Tomáš Pluskal [1] & Jing-Ke Weng [1,3]✉

Plant halogenated natural products are rare and harbor various interesting bioactivities, yet the biochemical basis for the involved halogenation chemistry is unknown. While a handful of Fe(II)- and 2-oxoglutarate-dependent halogenases (2ODHs) have been found to catalyze regioselective halogenation of unactivated C–H bonds in bacteria, they remain uncharacterized in the plant kingdom. Here, we report the discovery of dechloroacutumine halogenase (DAH) from Menispermaceae plants known to produce the tetracyclic chloroalkaloid (−)-acutumine. DAH is a 2ODH of plant origin and catalyzes the terminal chlorination step in the biosynthesis of (−)-acutumine. Phylogenetic analyses reveal that DAH evolved independently in Menispermaceae plants and in bacteria, illustrating an exemplary case of parallel evolution in specialized metabolism across domains of life. We show that at the presence of azide anion, DAH also exhibits promiscuous azidation activity against dechloroacutumine. This study opens avenues for expanding plant chemodiversity through halogenation and azidation biochemistry.

[1] Whitehead Institute for Biomedical Research, Cambridge, MA 02142, USA. [2] Department of Biological Engineering, Massachusetts Institute of Technology, Cambridge, MA 02139, USA. [3] Department of Biology, Massachusetts Institute of Technology, Cambridge, MA 02139, USA. ✉email: wengj@wi.mit.edu

Halogenated natural products are commonly found in bacteria, marine animals and macroalgae, and harbor a multitude of biological activities ranging from antibiotic to anticancer[1–3]. Despite >5000 characterized halogenated natural products to date, only a few examples have been identified in land plants[1,4,5]. Some known cases include maytansine, a cytotoxic chlorinated ansamacrolide isolated from the Ethiopian shrub *Maytenus serrata*[5,6], and 4-chloroindole-3-acetic acid, an auxin analog found in the seeds of some leguminous plants[4]. While maytansine was recently confirmed to be produced by endophytic bacteria[5], the synthesis of 4-chloroindole-3-acetic acid may also rely on symbiotic bacterial metabolism, as known classes of indole halogenases, which are well characterized in bacteria, are absent in sequenced legume genomes[7].

Another notable example is (−)-acutumine, a tetracyclic chloroalkaloid produced in plants of the Menispermaceae family. (−)-Acutumine exhibits selective cytotoxicity to cultured human T cells[8] and memory-enhancing properties in the Wistar rat model[9]. Due to its unusual structure and bioactivities, total synthesis of (−)-acutumine has been pursued by synthetic chemists in recent years[10–12]. In particular, (−)-acutumine contains an $sp^3$-carbon-halogen bond, which is difficult to install in a site-specific manner, especially at the late stages of chemical synthesis. Early studies have also explored (−)-acutumine biosynthesis in Menispermaceae plants using isotopic tracing showing that tyrosine is a biosynthetic precursor of (−)-acutumine and chlorination is the terminal step of its biosynthesis[13,14].

Regio-selective and stereo-selective halogenation is of great interest in medicinal chemistry, as the bioactivities and pharmacokinetics of lead pharmacophores can often be finetuned through halogenation[15]. Using bacterial halogenases, previous studies have introduced halide functional groups to vinca alkaloids in periwinkle plant *Catharanthus roseus*[16], and have also generated new-to-nature halogenated indigo precursors in *Nicotiana benthamiana*[17]. However, with the scarcity of naturally occurring halogenated plant metabolites, no halogenase of plant origin has been characterized to date. We therefore set out to investigate the chlorination biochemistry in (−)-acutumine biosynthesis in Menispermaceae plants.

Here, we report the discovery of dechloroacutumine halogenase (DAH) from Menispermaceae plants, an Fe(II)-dependent and 2-oxoglutarate-dependent halogenase (2ODH) that catalyzes the terminal chlorination reaction in (−)-acutumine biosynthesis. Phylogenetic analyses suggest that DAH evolved independently from those 2ODHs previously identified from bacteria[18–23]. We show that DAH can also catalyze regioselective C–H azidation at the presence of azide anion, illustrating the promising broader utilities of 2ODH enzymes for versatile natural product derivatization.

## Results and discussion

**Discovery of dechloroacutumine halogenase from Menispermaceae plants.** We obtained three species of Menispermaceae plants, two known (−)-acutumine producers *Menispermum canadense* and *Sinomenium acutum*, and one non-producer *Stephania japonica* (Fig. 1a). Using liquid chromatography high-resolution accurate-mass mass spectrometry (LC-HRAM-MS), we detected several previously reported acutumine-type alkaloids in *M. canadense* and *S. acutum*[13,14,24] (Fig. 1b). The same set of alkaloids were absent in *S. japonica* extracts (Fig. 1b), suggesting that comparative transcriptomic analyses including *S. japonica* as a reference species devoid of the (−)-acutumine chemotype could help identify genes involved in (−)-acutumine biosynthesis. Furthermore, we found that (−)-acutumine accumulation is

relatively more enriched in roots compared to stems and leafs in *M. canadense*.

To enable gene discovery in the target Menispermaceae plants, we performed tissue-specific RNA-Seq experiments, followed by de novo transcriptome assembly for each of the three Menispermaceae species. The resulting transcripts were annotated by BLAST search against the UniProtKB/Swiss-Prot database, and their tissue-specific abundance was quantified by transcripts per million (TPM) values. Using $D_2O$-labeling technique, a previous study explored the plausible (−)-acutumine biosynthetic pathway by tracing deuterium-labeled benzylisoquinoline alkaloids (BIAs) in *M. canadense*[25]. Based on this proposed pathway and prior knowledge about BIA metabolism in other plants, we examined functionally annotated transcripts that are highly and differentially expressed in the root versus the leaf or stem tissue of *M. canadense*, from which numerous candidate genes for the biosynthesis of (−)-acutumine and other related BIAs were identified (Fig. 1c and Supplementary Figs. 1, 2; Supplementary Table 1, 2). We postulated that the candidate enzyme responsible for the terminal chlorination is catalyzed by 2ODH because it is the only characterized class of halogenases capable of targeting unactivated $sp^3$-hybridized carbon centers for single-step halogenation[26]. We therefore sought highly expressed candidate genes in our transcriptomes that are annotated as Fe(II)-dependent and 2-oxoglutarate-dependent dioxygenases (2ODDs), and manually searched for the potential presence of active-site sequence variation unique to the 2ODH subfamily.

2ODHs were previously identified in bacteria, such as SyrB2 from syringomycin E biosynthesis in *Pseudomonas syringae*[18], WelO5 from welwitindolinone biosynthesis in *Hapalosiphon welwischii*[26], and recently reported amino acid halogenases from *Streptomyces cattleya*[20]. 2ODHs are evolutionarily derived from the 2ODDs, a large enzyme family spanning all kingdoms of life[27]. Canonical 2ODDs activate molecular oxygen using a ferrous cofactor and a 2-oxoglutarate (2OG) cosubstrate to oxidize aliphatic C–H bonds at a highly conserved iron-binding facial triad $(HxD/EX_nH)$[28], typically resulting in a hydroxylation outcome. In contrast, bacterial 2ODHs harbor a mechanistic active-site substitution, replacing the key acidic residue Asp or Glu of the facial triad in 2ODDs with a Gly or Ala. This allows the halide ligand to occupy an iron coordination site, and in turn elicits halogenation chemistry[26]. With this knowledge in mind, we discovered that the most highly expressed 2ODD-family gene in *M. canadense* root transcriptome is likely to encode a candidate 2ODH, because it harbors the unusual active-site Asp226Gly substitution similar to bacterial 2ODHs (Fig. 1c, d and Supplementary Fig. 2, Supplementary Dataset 1). This gene also exhibits a 459-fold higher transcript expression level in roots than in leaves, consistent with the more enriched (−)-acutumine accumulation in roots. This gene is named as *M. canadense dechloroacutumine halogenase* (*McDAH*) hereafter. Analysis of the *S. acutum* transcriptome identified SaDAH, which exhibits 99.1% sequence identity to McDAH at the protein level, and is likely an ortholog of McDAH (Fig. 1d and Supplementary Fig. 3). No obvious DAH ortholog could be identified in the *S. japonica* transcriptome, and no other annotated 2ODD amongst the three transcriptomes harbor the Asp-to-Gly facial triad mutation. Moreover, no homologs of the other aforementioned classes of halogenases could be identified in the three Menispermaceae plant transcriptomes.

**Biochemical characterization of dechloroacutumine halogenase.** To examine the biochemical function of DAH, we first expressed and purified recombinant SaDAH protein from *Escherichia coli* (Supplementary Fig. 4), and performed in vitro

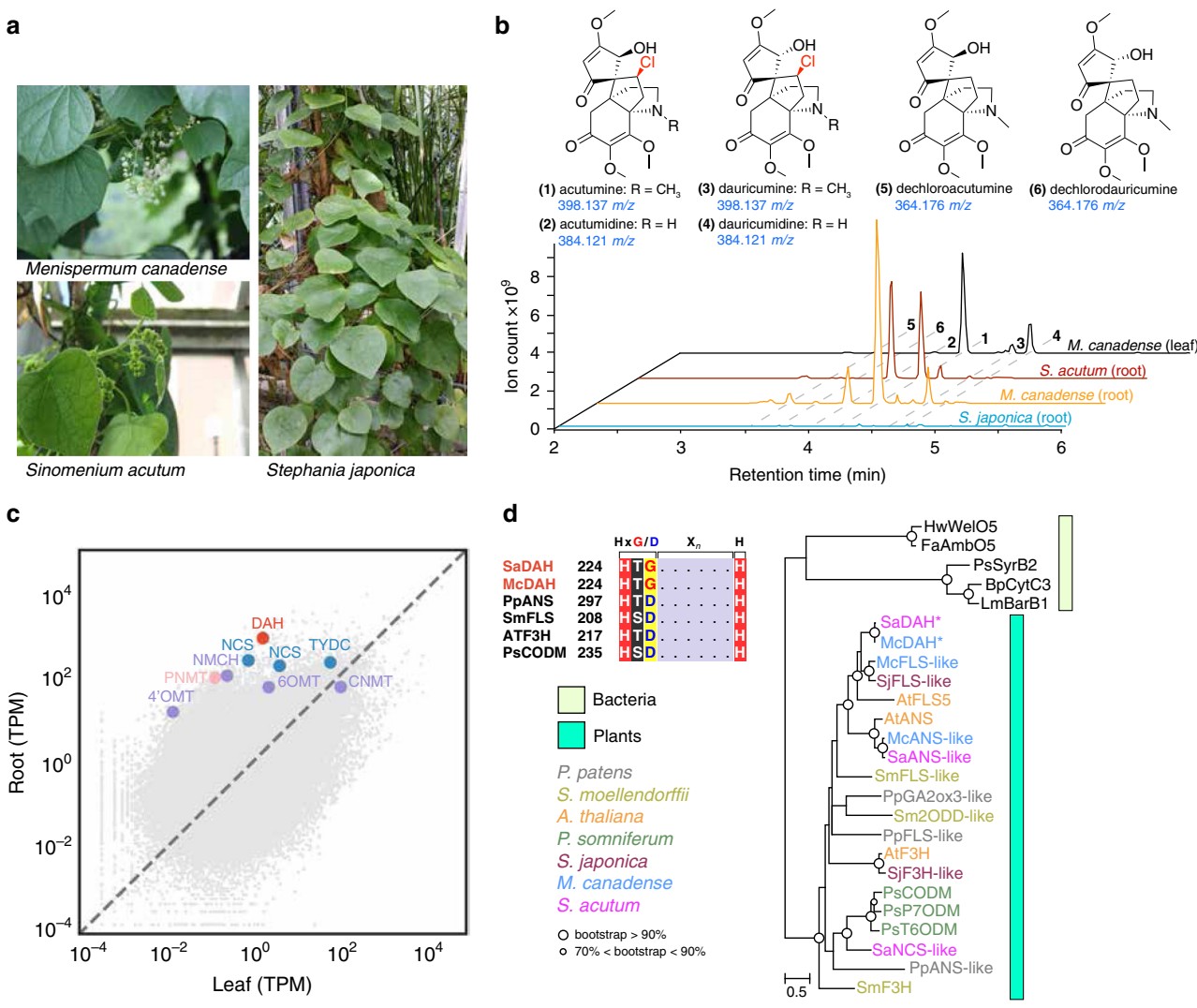

**Fig. 1 Identification of candidate genes encoding DAH in Menispermaceae plants. a** Images of *M. canadense*, *S. acutum*, and *S. japonica* plants. **b** Extracted ion chromatograms (XICs) exhibiting the presence and absence of (−)-acutumine and its related compounds in three Menispermaceae plants. **c** Transcript differential expression analysis in *M. canadense* root vs. leaf tissues. Transcript abundance is quantified by transcripts per million (TPM) values. Candidate (−)-acutumine biosynthetic enzymes are colored based on their postulated roles in the proposed pathway depicted in Supplementary Fig. 1. **d** Sequence and phylogenetic analysis of DAH. The insert at the upper left corner shows multiple sequence alignment of two DAHs and other ODDs highlighting the Hx (D/G)X$_n$H motif. The full sequence alignment is shown in Supplementary Fig. 3. The phylogenetic tree of DAHs together with select 2ODD-family proteins is inferred using maximum-likelihood method. Bootstrap statistics (200 replicates) are indicated at the tree nodes. The scale measures evolutionary distance in substitutions per amino acid. Asterisks denote the two orthologous DAHs identified in this study. The multiple sequence alignment used for building the phylogenetic tree is in Supplementary File 1. NCS norcoclaurine synthase, TYDC tyrosine decarboxylase, PNMT pavine N-methyltransferase, 6OMT (*RS*)-norcoclaurine 6-*O*-methyltransferase, CNMT (*S*)-coclaurine *N*-methyltransferase, NMCH (*S*)-*N*-methylcoclaurine-3-hydroxylase, 4'OMT (*S*)-3'-hydroxy-N-methylcoclaurine 4'-*O*-methyltransferase, ANS anthocyanidin synthase, FLS flavonol synthase, F3H flavanone-3-hydroxylase, CODM codeine *O*-demethylase, GA2ox3 gibberellin 2-beta-dioxygenase 3, 2ODD iron(II)- and 2-oxoglutarate-dependent dioxygenase, P7ODM papaverine 7-*O*-demethylase, T6ODM thebaine 6-*O*-demethylase.

enzyme assays. SaDAH exhibits specific 2OG-dependent halogenase activity that converts (−)-dechloroacutumine to (−)-acutumine in a reaction buffer containing 1 mM NaCl (Fig. 2a, b and Supplementary Fig. 5), whereas no halogenase activity could be detected when the structurally related BIA sinomenine was tested as a substrate (Supplementary Fig. 6). Recombinant McDAH exhibits identical activity to SaDAH (Supplementary Fig. 7). Since the recombinant protein yield was higher for SaDAH than McDAH, we chose SaDAH as the representative DAH for detailed characterizations hereafter. To examine the potential substrate promiscuity of SaDAH, we tested its halogenase activity against a panel of additional alkaloids including vinpocetine, codeine, scoulerine, berberine, and boldine. No halogenated products could be detected in these assays, suggesting that SaDAH is likely a specific halogenase towards its native substrate (−)-dechloroacutumine (Supplementary Fig. 8).

SaDAH exhibits a $K_M$ of 18.35 ± 7.13 μM against (−)-dechloroacutumine as the substrate, and an apparent $k_{cat}$ of 63.44 ± 8.75 min$^{-1}$ (Supplementary Fig. 9), which are comparable to the kinetic constants measured for previously characterized bacterial 2ODHs[20] and plant 2ODDs[29]. Interestingly, a trace amount of the hydroxylation product 11-hydroxy-dechloroacutumine was also detected in the SaDAH in vitro enzyme assay (Supplementary Fig. 10), suggesting that SaDAH still retains a low level of

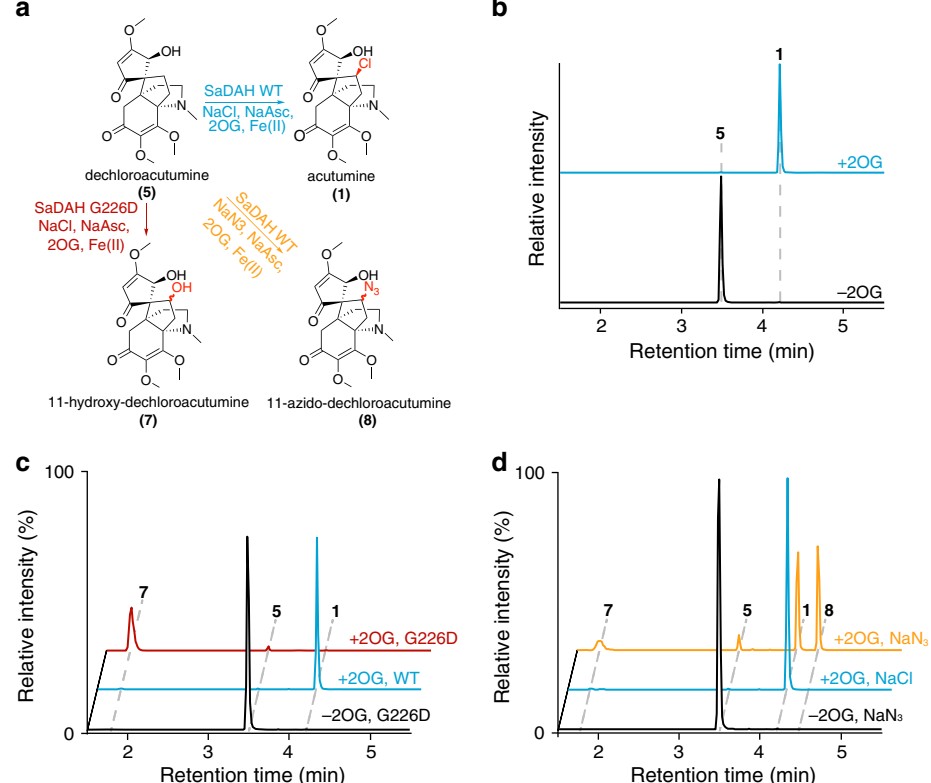

**Fig. 2 Functional characterization of DAH and its G226D mutant by in vitro enzyme assays. a** Alternative catalytic activities of SaDAH-WT and SaDAH-G226D. The chlorination, hydroxylation and azidation activities are colored in blue, red, and yellow, respectively. The putative regiochemistry of the −OH and −N$_3$ in **7** and **8** are highlighted in red, while the stereochemistry of the products remains undetermined. **b** XICs showing the in vitro activity of SaDAH-WT that chlorinates **5** to yield **1** in a 2OG-dependent manner. **c** XICs showing the in vitro activity of SaDAH-G226D that predominantly hydroxylates **5**–**7** in a 2OG-dependent manner. The chlorination product **1** is also produced as a minor product. **d** XICs showing the 2OG-dependent azidation activity of SaDAH-WT that converts **5**–**8** in reaction buffer containing 1 mM NaN$_3$. The chlorination product **1** and the hydroxylation product **7** are also produced as side products. Mass windows used for displaying the XICs: **5**, 364.176 $m/z$; **1**, 398.137 $m/z$; **7**, 380.17 $m/z$; and **8**, 405.18 $m/z$.

latent hydroxylase activity, although the hydroxylation product was not detected in any of the Menispermaceae plant extracts. This retention of ancestral hydroxylation activity has also been observed in the previously characterized 2ODHs[26].

**Independent evolution of 2ODHs in plants and bacteria**. To the best of our knowledge, our discovery of DAH in the context of (−)-acutumine biosynthesis represents the first 2ODH of plant origin. Remarkably, DAH seems to harbor the same active-site Asp226Gly substitution previously shown to be characteristic of bacterial 2ODHs[26]. To discern whether DAH might have evolved from ancestral plant 2ODDs, or, alternatively, through a plausible horizontal gene transfer event from bacteria, we performed maximum-likelihood phylogenetic analysis of DAHs together with bacterial 2ODHs and other 2ODD-family proteins from Menispermaceae plants, *Arabidopsis thaliana*, *Papaver somniferum*, *Selaginella moellendorffii*, and *Physcomitrella patens*. DAHs appear to be immediately sister to the well-characterized plant flavonol synthases (FLSs), whereas bacterial 2ODHs form a distinct clade separate from all plant 2ODD-family proteins (Fig. 1d and Supplementary Data 1). Moreover, an extended phylogenetic analysis including 2ODD homologs identified from the available transcriptomes of Ranunculales plants further supports that DAH recently evolved from an FLS-like progenitor through a gene duplication event that occured in during the radiation of Ranunculales, contributing to the occurence of (−)-acutumine biosynthesis in Menispermaceae (Supplementary Fig. 11, Supplementary Data 2).

**Expansion of C–H functionalization using dechloroacutumine halogenase**. To assess the role of the potential mechanistic Asp226Gly substitution found in DAHs relative to canonical 2ODDs, we generated the SaDAH G226D mutant and examined its in vitro enzymatic activity. The SaDAH G226D mutant is entirely devoid of halogenase activity, and instead exhibits exclusive (−)-dechloroacutumine hydroxylase activity to give the hydroxylation product 11-hydroxy-dechloroacutumine (Fig. 2a, c and Supplementary Fig. 12). This result indicates that DAH adopts the same mechanistic substitution as in bacterial 2ODHs, wherein the canonical active-site iron-coordinating Asp is replaced with an Gly or Ala that facilitates Cl$^-$ binding and the subsequent chlorination chemistry[26]. As predicted by this mechanism, mutating Gly226 of DAH to Asp therefore restores the canonical 2ODD iron-binding facial triad, prevents halide coordination, and consequently reverts the reaction outcome to hydroxylation[26]. Interestingly, after an exhaustive search and examination of the abundant plant 2ODD-family protein sequences identifiable at NCBI, Uniprot, and 1KP databases, the two DAHs identified in this study remain the only plant 2ODD-family proteins that harbor the unusual Asp226Gly substitution, suggesting the occurrence of 2ODH activity through this particular molecular mechanism is a rare serendipity in plant metabolic evolution.

It was previously shown that bacterial halogenase SyrB2 exhibits extended catalytic activities to install alternative anions[30]. We therefore tested SaDAH's ability to derivatize (−)-dechloroacutumine using a panel of alternative anions: N$_3^-$, Br$^-$, NO$_3^-$,

$NO_2^-$, and $OCN^-$. Among the anions tested, only $N_3^-$ resulted in a derivatized product, which was identified as 11-azido-dechloroacutumine by LC-HRAM-MS analysis (Fig. 2a, d and Supplementary Figs. 12, 13). (−)-Acutumine and 11-hydroxy-dechloroacutumine were also produced in the same assay, likely due to the presence of trace $Cl^-$ carried over from the enzyme storage buffer and elevated hydroxylation outcome when enzyme access to native $Cl^-$ substrate is limited, respectively (Fig. 2d). The regio-specificity and stereo-specificity of the −OH and −N₃ moieties is not directly determined, but can be reasonably postulated to be at the C11 position of (−)-dechloroacutumine with the same stereochemistry as the −Cl group in (−)-acutumine (Fig. 2a, Supplementary Fig. 12). The fact that only a single −Cl product is observed in SaDAH chlorination assay indicates that the enzyme is not capable of abstracting an alternate C–H bond (Fig. 2b and Supplementary Fig. 5). This postulation is consistent with the nature of substrate binding and C–H bond abstraction previously reported in other bacterial 2ODHs[26].

In summary, we report the discovery of the first 2ODH of plant origin, DAH, that catalyzes the terminal chlorination step in (−)-acutumine biosynthesis in Menispermaceae plants. The independent occurrences of 2ODHs in plant and bacteria through the same key active-site mechanistic substitution illustrate an exemplary case of parallel evolution in specialized metabolism across domains of life[31]. Although it is unknown why halogenated natural products are so scarce in plants compared to those found in bacteria, fungi, and marine organisms, the lack of access to halogen salts could be one of the explanations. Future experiments to knockdown expression of *DAH* and other candidate BIA metabolic enzyme-encoding genes in Menispermaceae plants not only will help elucidate the (−)-acutumine biosynthetic pathway, but also will shed light on the functional significance of (−)-acutumine biosynthesis to its native plant hosts. In addition to chlorination, we show that DAH is also capable of catalyzing azidation chemistry, and thus opens up several new avenues for future applications. Azidation of plant natural products could enable rapid metabolite detection using click-chemistry-based fluorogenic probes[32,33]. The same chemical principle can also be applied to pull-down experiments for identifying unknown protein targets of specific bioactive plant natural products amenable to site-specific azidation. Moreover, as a rare 2ODH of plant origin, DAH presents a starting point for future directed evolution experiments to further expand substrate specificity and anion selectivity, which will find use in the production of new-to-nature plant natural products through the means of metabolic engineering[34].

## Methods

**Plant materials and chemicals**. *S. acutum* and *S. japonica* plant samples were collected at the University of Connecticut's EEB Biodiversity Education & Research Greenhouses, Connecticut. *M. canadense* plant samples were purchased from the Toadshade Wildflower Farm, New Jersey. All chemical reagents were obtained from Sigma-Aldrich or Thermo Fisher Scientific, unless otherwise stated. Ultrapure water was generated by a Milli-Q system (EMD Millipore). (−)-Acutumine (>98%) was obtained from BOC Sciences and as a gift from the Herzon Lab at Yale University. Sinomenine, scoulerine, and vinpocetine were obtained from Santa Cruz Biotechnology.

**Plant metabolite extraction**. Approximately 10 mg of plant tissue was dissected, weighed, transferred into grinding tubes containing approximately six zirconia/silica disruption beads (2 mm diameter; Research Products International), and flash-frozen in liquid nitrogen. The frozen samples were homogenized on a TissueLyser II (QIAGEN) at $25 s^{-1}$ for 2 min. Metabolites were extracted using 100 volumes (w/v) of 80% MeOH at 65 °C for 10 min. Extracts were centrifuged once ($13,000 \times g$, 5 min) to pellet plant tissue, and the supernatants were filtered using 0.2 μm filter vials (Thomson Instrument Company) for LC-HRAM-MS analysis.

**Transcriptome sequencing and assembly**. Total RNA was extracted using the RNeasy Mini Kit (Qiagen) from the leaf, stem, and root of *Menispermum*

*canadense* (3–6 biological replicates each), from the root of *Sinomenium acutum* (2 biological replicates), and from the root of *Stephania japonica* (2 biological replicates). RNA quality was assessed by Bioanalyzer (Agilent Technologies). Strand-specific mRNA libraries were prepared using the KAPA Hyper Prep mRNA Library Prep Kit (Kapa Biosystems), and sequenced on a HiSeq 2500 sequencer (Illumina) in paired-end mode (PE100). Total 342 M, 49 M, and 50 M paired sequencing reads were acquired for *M. canadense*, *S. acutum*, and *S. japonica*, respectively. Sequence FASTQ files were trimmed for sequencing adapters and poor-quality reads using Trimmomatic (v. 0.39) and assembled into de novo transcriptomes using Trinity (v. 2.8.5) in strand-specific mode[35,36]. Putative open reading frames in transcripts were identified using Transdecoder (v. 5.5.0)[37]. Gene expression transcripts-per-million (TPM) statistics were determined using kallisto (v. 0.46.0)[38]. Transcripts and predicted protein sequences were annotated with TPM values and closest BLAST hits from the UniProtKB/Swiss-Prot database using in-house scripts. Transcriptome mining was performed on a local BLAST server[39]. To compare the RNA expression profiles of *M. canadense*, *S. acutum*, and *S. japonica*, orthologs were identified by clustering their geneset peptides using the OrthoFinder (v.2.2.7)[40] pipeline with default parameters.

**Organic synthesis of (−)-acutumine**. The reaction schematic is shown in Supplementary Fig. 14. All reagents were used as supplied without further purification with the exception of AIBN, which was recrystallized from acetone and dried over $P_2O_5$. Analytical TLC was conducted using Millipore $SiO_2$ 60 F254 TLC (0.250 mm) plates. Mass spectrometric analysis was performed on a TSQ Quantum Access Max mass spectrometer (Thermo Fisher Scientific) operated in positive ionization mode with a full scan range of 100–400 *m/z*. A solution of (−)-acutumine (6.0 mg, 0.015 mmol) in dry toluene (6.0 mL) at 23 °C was treated successively with ⁿBu₃SnH (0.10 mL, 0.38 mmol) and AIBN (1.3 mg, 0.008 mmol) under a nitrogen atmosphere. The resulting solution was refluxed for 3 h, then cooled to 23 °C and treated with saturated NaHCO₃(aq) (15 mL) and sat. KF(aq) (1 mL). The organic layer was separated and the aqueous layer was extracted with CHCl₃ (2 × 10 mL). The combined organic extracts were dried over Na₂SO₄ and concentrated on a rotary evaporator. The resulting solid residue was purified by preparative reverse-phase HPLC (Shimadzu Preparative HPLC with LC-20AP pump and SPD-20A UV–VIS detector) using Kinetex 5μ C₁₈ 100 A, 150 × 21.2 mm column, solvent A-(H₂O-0.1% TFA), solvent B-(CH₃CN-0.1% TFA), 0–50 min 40–80% B, 10 mL/min) to provide (−)-dechloroacutumine[8] as an amorphous white solid (30 μg, 1%). Spectrometric data are in agreement with the values published previously[11].

**Recombinant protein expression and purification**. The *SaDAH* gene was codon-optimized for heterologous overexpression in *Escherichia coli*, and purchased as a synthetic gene (Integrated DNA Technologies). The *SaDAH* gene was cloned into pHis8-4 expression vector containing an *N*-terminal 8× His tag followed by a tobacco etch virus (TEV) cleavage site (Supplementary Table 3). The verified pHis8-4b::*SaDAH* construct was transformed into *E. coli* BL21 (DE3) for recombinant protein expression. A 1 L culture of terrific broth (TB) medium containing 50 μg/mL kanamycin was inoculated with 30 mL of an overnight starter culture and allowed to grow with shaking at 200 rpm at 37 °C to an OD₆₀₀ of 0.6–0.8. Then protein expression was induced by addition of 0.5 mM isopropyl β-D-1-thiogalactopyranoside (IPTG) followed by cold shock of the medium and subsequent growth with shaking at 200 rpm (18 °C for 18 h).

Cultures were harvested by centrifugation and the resulting cell paste (~10 g/L) was resuspended in lysis buffer (100 mM Tris pH 8.0, 200 mM NaCl, 20 mM imidazole, 10% (vol/vol) glycerol, 1 mM dithiothreitol) containing 1 mg/mL lysozyme and 1 mM phenylmethylsulfonyl fluoride. Cells were lysed via five passes through an M-110L microfluidizer (Microfluidics). The resulting crude protein lysate was clarified by centrifugation ($19,000 \times g$, 45 min) before QIAGEN nickel-nitrilotriacetic acid (Ni-NTA) gravity flow chromatographic purification. After loading the clarified lysate, the Ni-NTA resin was washed with 20 column volumes of lysis buffer and eluted with 1 column volume of elution buffer (100 mM Tris pH 8.0, 200 mM NaCl, 250 mM imidazole, 10% (vol/vol) glycerol, 1 mM dithiothreitol). Then 1 mg of His-tagged TEV protease was added to the eluted protein, followed by dialysis at 4 °C for 16 h in dialysis buffer (25 mM Tris pH 8.0, 200 mM NaCl, 5% (vol/vol) glycerol, 5 mM EDTA, 0.5 mM dithiothreitol). After dialysis, protein solution was passed through Ni-NTA resin to remove uncleaved protein and His-tagged TEV. The recombinant proteins were further purified by gel filtration on an ÄKTA Pure fast protein liquid chromatography (FPLC) system (GE Healthcare Life Sciences). The principal peaks were collected, verified by SDS polyacrylamide gel electrophoresis and dialyzed into a storage buffer (25 mM Tris pH 8.0, 5%(vol/vol) glycerol). Finally, proteins were concentrated to >10 mg/mL using Amicon Ultra-15 Centrifugal Filters (Millipore).

***SaDAH* mutagenesis**. *SaDAH* G226D mutant was generated according to the protocol described in the QuikChange site-directed mutagenesis kit (Agilent Technologies) using plasmid pHis8-4::*SaDAH* as the template and the following primer sequences:

SaDAH-G226D-5′: GTGTAGTCCCTCATACAGACTATCCTGCAATGAC AAT; SaDAH-G226D-3′: CACATCAGGGAGTATGTCTGATAGGACGTT ACTGTTA (Supplementary Table 4). The resulting pHis8-4::Sa*DAH-G226D*

construct was verified by sequencing. Recombinant mutant protein production and purification were carried out following the same procedure as described above.

**In vitro enzyme activity assays.** Each enzyme assay for SaDAH-WT and SaDAH-G226D with (−)-dechloroacutumine was carried out in 25 mM Tris buffer, pH = 8.0, on a 50-μL scale containing the following components: enzyme (20 μM), (−)-dechloroacutumine (100 μM), 2OG (500 μM), NaCl (1 mM), sodium ascorbate (5 mM), and $(NH_4)_2Fe(So_4)_2$ (2 mM). In a typical assay, the components were added in the following order: (1) Tris, (2) NaCl, (3) enzyme, (4) sodium ascorbate, (5) 2OG, (6) (−)-dechloroacutumine, and (7) $(NH_4)_2Fe(So_4)_2$. The samples were incubated under aerobic conditions at room temperature for 1 h. The assays were quenched by adding methanol to 50% final concentration and centrifugation to spin down enzymes and debris. The supernatants were dried under nitrogen gas and resuspended in 10% acetonitrile in water for injection into the LC-HRAM-MS system for analysis. The azidation activity assay of SaDAH-WT was carried out following the procedure described above, with the replacement of NaCl with $NaN_3$ (1 mM).

**SaDAH enzyme kinetic assays.** Each enzyme reaction (50 μL) contained SaDAH-WT (30 nM), 2OG (500 μM), NaCl (1 mM), sodium ascorbate (5 mM), $(NH_4)_2Fe$ $(So_4)_2$ (2 mM), and (−)-dechloroacutumine (0–105 μM). Reactions were initiated by addition of a mixture containing SaDAH-WT and $(NH_4)_2Fe(So_4)_2$ in the presence of varying concentrations of (−)-dechloroacutumine (0–105 μM; 18 different concentrations). Each reaction was quenched after 10 min by adding methanol to 50% final concentration and centrifugation to spin down enzymes and debris. The supernatants were dried under nitrogen gas and resuspended in 10% acetonitrile in water for injection into the LC-HRAM-MS system for analysis. The reaction velocity for each assay was measured by integrating the peak area corresponding to (−)-acutumine on LC-MS. An internal standard curve of (−)-acutumine was generated using the same set of reactions with known concentrations of (−)-dechloroacutumine (0–105 μM), but quenching each reaction at 3 h after the substrate was completely converted to (−)-acutumine.

**LC-MS analysis.** LC was conducted on a Dionex UltiMate 3000 UHPLC system (Thermo Fisher Scientific) using water with 0.1% formic acid as solvent A and acetonitrile with 0.1% formic acid as solvent B. Reverse phase separation of analytes was performed on a Kinetex C18 column, $150 \times 3$ $mm^2$, 2.6 μm particle size (Phenomenex). The column oven was held at 30 °C. Most injections were eluted with 5% B for 0.5 min, a gradient of 5–36% B for 5 min, 95% B for 2 min and 5% B for 2.5 min, with a flow rate of 0.5 mL/min. The analysis of Menispermaceae species tissue extracts and enzyme substrate promiscuity tests was performed with 5% B for 0.5 min, a gradient of 5–95% for 14.5 min, 95% B for 2 min and 5% B for 3 min, with a flow rate of 0.5 mL/min. Most MS analyses were performed on a high-resolution Q-Exactive benchtop Orbitrap mass spectrometer (Thermo Fisher Scientific) operated in positive ionization mode with full scan range of 100–600 $m/z$ and top five data-dependent MS/MS scans. Initial enzyme activity assays were analyzed with a TSQ Quantum Access Max mass spectrometer (Thermo Fisher Scientific) using multiple reaction monitoring at $m/z$ transitions corresponding to (−)-acutumine (398.14–273.08, 305.10, 341.08) and (−)-dechloroacutumine (364.18–257.08, 289.11, 307.12) with scan width of 0.5 and collision energy of 20 V. For detection of (−)-acutumine derivatives, single ion monitoring was used to scan the following $m/z$ values: 364.20, 380.17, 398.1, 405.17, 409.16, 425.16, and 442.08 with scan width of 0.5 and collision energy of 20 V. The MS analysis for DAH substrate promiscuity tests was conducted on the TSQ Quantum Access Max mass spectrometer (Thermo Fisher Scientific) operated in positive ionization mode with a full scan range of 250–500 $m/z$. Raw LC-MS data were analysed using XCalibur (Thermo Fisher Scientific).

**Sequence alignment and phylogenetic tree analysis.** The sequences for sequence alignment and phylogenetic tree construction were obtained from the NCBI database, 1KP database, Phytozome v12.1.6. pBLAST with DAH as a query sequence was conducted and top 20 full length protein sequences were collected for each species: *P. patens*, *S. moellendorffii*, *A. thaliana*, all available species in Ranunculales plant family, *S. japonica*, *M. canadense*, and *S. acutum*. In-house scripts were used to extract sequences from the assembled transcriptomes discussed in this manuscript. Sequence alignments were performed using the MUSCLE[41] algorithm in MEGA7[42] and visualized using ESPript 3[43]. Evolutionary histories were inferred by using the maximum-likelihood method on the basis of the JTT matrix-based model. Bootstrap statistics were calculated using 200 replicates. All phylogenetic analyses were conducted in MEGA7[42]. All alignment files can be found under Associated Data.

## Data availability

The sequences of the genes reported in this article have been deposited in NCBI GenBank (accessions MT040708, MT040709). Protein expression plasmids used in this study are available from Addgene (accessions 140128-140130). The raw sequencing reads have been submitted to NCBI SRA (accessions SRR10947794-SRR10947810) and the de novo assembled transcriptome to NCBI TSA (accessions GIIU00000000.1,

GIIT00000000.1, GIIS00000000.1). The raw data underlying Supplementary Figs. 4, 9 are provided as a Source Data file. Additional data that support the findings of this study are available from the corresponding authors upon reasonable request.

## Code availability

Codes used to assemble the transcriptomes and extract sequences discussed in this manuscript are available on request.

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

## Acknowledgements

This work was supported by grants from the Beckman Foundation (J.K.W.), the Keck Foundation (J.K.W.), and the National Science Foundation (CHE-1709616). We thank the Herzon Lab at Yale University for providing (−)-acutumine, and the EEB Biodiversity Education & Research Greenhouses at University of Connecticut for providing the *S. acutum* and *S. japonica* plants used in this study, along with their images. We also thank Lady Bird Johnson Wildflower Center for providing the *M. canadense* image.

## Author contributions

C.Y.K. performed most experiments. A.J.M. and T.P. performed the RNA-seq experiment and helped assemble the Menispermaceae plant transcriptomes. F.L. and C.M.G. helped synthesize and purify (−)-dechloroacutumine used for enzyme assays. C.Y.K. and J.K.W. designed experiments, analyzed data, and wrote the manuscript. All authors reviewed the manuscript.

## Competing interests

J.K.W. is a co-founder, a member of the Scientific Advisory Board, and a shareholder of DoubleRainbow Biosciences, which develops biotechnologies related to natural products. All other authors have no competing interests.
