## [Peer Review File · Nature Communications]

Reviewers' comments:

Reviewer #1 (Remarks to the Author):

The authors describe the discovery of a 2-oxoglutarate/Fe(II)-dependent halogenase of plant origin. The identified enzyme, named dechloroacutumine halogenase (DAH), catalyzes the terminal chlorination step in the biosynthesis of the chloroalkaloid (-)-acutumine in plants of the Menispermaceae family. The corresponding genes encoding DAH in *Sinomenium acutum* and *Menispermum canadense* were identified by comparative transcriptomic analyses versus the non-producer *Stephania japonica*, by tissue specific analyses (root vs. leaf tissues) of all three plant species and by probing for iron(II)- and 2-oxoglutarate-dependent dioxygenases.

Ordering synthetic genes for the identified enzymes, the authors were able to express functional DAH enzymes from *S. acutum* and *M. canadense* in *E. coli* and could show halogenation activity in vitro. Furthermore, the authors highlighted anion promiscuity of the enzymes SaDAH and gave evidence that alternative catalytic activities can be accessed by SaDAH and its variants (azidation and hydroxylation).

The research is well described and clearly presented and the manuscript's findings are a valuable contribution to the field as the discovery of a freestanding 2-oxoglutarate/Fe(II)-dependent halogenases of plant origin are expanding the (so far very small) biocatalytic toolbox of these type of halogenases.

However, prior to publication in *Nature Communications*, there are a number of concerns which need to be addressed:

Major points

(1) The authors show that the newly discovered enzymes exhibit a certain anion promiscuity such as for NaN₃. To increase the depth of the study, it would be interesting to also learn more about the enzymes' substrate promiscuity, for example with respect to related alkaloids such as hasubanonine.

(2) The depicted LCMS analyses are extracted ion chromatograms. It would be important to also show a TIC of the in vitro chlorination to rule out other major side products.

(3) Supplementary information / Supplementary Figure 11: Dechlorination of (-)-acutumine: Recorded spectrometric data should be shown, purity of obtained product should be described.

(4) The authors should give a more detailed description in the main text how the proposed (-)-acutumine biosynthetic pathway was elucidated (Figure S1). Did the differential expression analysis in fact only yield one possible enzyme sequence candidate for each biosynthetic step as suggested in Figure 1c and S2?

(5) Fe(II)-dependent halogenases have been previously described to exhibit hydroxylation activity (as also observed by the authors for the DAH enzymes). Importantly however, regioselectivity of CH-activation may differ depending on the reaction chemistry (e.g. hydroxylation vs. halogenation) as selectivity is determined by the orientation of the substrate with respect to the HO-Fe-Cl (Mathews et al., *PNAS* 2009, 106, 17723-17728; Hayashi et al., *ANIE* 2019, 58, 18535-18539). Whereas the authors do have an acutumine standard to which they compare the obtained MS and MS₂ data, such a standard is not described for the azide and hydroxylated (side) products. Nevertheless, the authors postulate a structure for the obtained hydroxy and azido derivatives of acutumine (Figure 2a, Figure S9). Contrary to the author's argument in the Figure legend S9, in light of the findings of above mentioned papers structural data is required to verify regio- and stereospecificity of the proposed products.

(6) What is the total turnover number of the DAH enzymes?

(7) Is there any evidence for the proposed gene duplication event?

(8) The authors conducted transcriptome analysis of all three plant species but only in case of *M. canadense*, this analysis was performed on different plant organs (root, leaf, stem). Does the plant *S. acutum* also show a difference in acutumine production in roots versus leaf?

(9) The authors should state more clearly, why SaDAH was chosen as the representative DAH for further analysis.

Other (minor) points:

(1) Page 2: "To identify the candidate enzyme responsible for the terminal chlorination, we probed several known classes of halogenases, including heme haloperoxidases¹⁶, vanadium-dependent haloperoxidases²⁰, flavin-dependent halogenases¹⁸, S-adenosyl-L-Met-dependent halogenases²¹, methyl halide transferases¹⁶, and iron(II)- and 2-oxoglutarate-dependent halogenases (2ODHs)²²."

This sentence is confusing. The authors should describe shortly in which way they "probed" the obtained transcripts for all of the enumerated enzyme classes.

(2) Supplementary information: Protein expression and purification

The authors should include a reference SDS-Page gel image to show purity of the wildtype DAH halogenases and the constructed variants.

(3) Supplementary information: SaDAH-WT kinetic assay

How was the internal standard curve (obtained by quenching of enzymatic reaction after 3h hours) verified? Did the authors compare with integrated peak areas of product solutions with known concentrations to ensure that conversion of substrate to product was indeed quantitative?

(4) Supplementary Figure 2 depicts a differential expression analysis of *M. canadense* root versus stem transcriptomes. What is the difference / interpretation of this data compared to the data presented in the main article (root versus leaf)?

(5) Page 3: The name of the species should be written out with first mentioning (*Menispermum canadense*)

(6) The following sentence needs to be clarified, as in DAH enzymes alanine corresponds to the wildtype.

"As predicted by this mechanism, mutating Ala226 back to Asp readily converts DAH to a hydroxylase"

Reviewer #2 (Remarks to the Author):

This is an excellent paper describing well executed research that led to the isolation and partial characterization of a truly novel 2-oxoglutarate- and iron-dependent dioxygenase that catalyzes the chlorination of an alkaloid substrate as the final catalytic step in the formation of the unique chloroalkaloid (-)-acutumine in plants of the Menispermaceae family. The strategy to identify gene candidates and the biochemical characterization of the isolated enzyme are clever and scientifically sound, respectively. The plant biology aspects of the work are generally limited to a comparative transcriptional analysis of the newly isolated halogenase, which displays a similar root- versus stem-enhanced expression compared with uncharacterized orthologs putatively encoding other alkaloid biosynthetic enzymes. Although most of these orthologs are likely to encode the proposed enzymes, this caveat should be better defined in the manuscript. It would clearly be nice to include

a gene knockdown-type experiment to further confirm the physiological relevance of the enzyme, but this is clearly and generally difficult in non-model plant systems.

From a biochemical perspective, it would be interesting to include empirical data showing the substrate specificity, or lack thereof, of the novel halogenase. Other basic aspects related to the characterization of the enzyme are strong, and the discussion is pertinent and informative.

Reviewer #3 (Remarks to the Author):

The authors reported the discovery of first enzyme of plant origin involved in halogenation, which is rare and would be an important biotechnological tool for expanding chemodiversity of plant natural products. I suppose that the impact and novelty of the finding reach the standard of the journal and are presented well generally. Only concern of mine is the amount of data, which seem fair for brief, but not for full, article. I hope that additional experiments suggested below would help to improve the study in that direction.

1, They showed that SaDAH-G226H catalyzes the hydroxylation rather than halogenation. Is it plausible to confer the halogenation activity to related ODDs with a substitution HxG? This issue is of great interest, considering the generation of novel halogenation catalysts.

2, In planta experimental evidences are lacking at all. Better to add the experiments technically possible to make the study more sound and lengthy enough, I believe.

3, It would be better to deliver a reason (more enriched (-)-acutumine accumulation in the roots) before mentioning the selection of candidate transcripts based on their high expression in the roots.

Reviewer #4 (Remarks to the Author):

The work by Kim et al targets a rare yet important plant secondary metabolic pathway. Halogenated organic compound are frequent in pharmaceuticals and many other synthetic organic compounds, but are rarely found in plants. The authors targeted one these rare plant groups and used a simple yet elegant approach to identify candidate genes. They combined a simple comparative root versus leaf transcriptomics approach with candidate gene family selection and biochemical rational to identify a single candidate gene to catalyze the final chlorination reaction to produce the halogenated compound, called acutumine. Heterologous expression of the candidate protein in *E. coli* validated the predicted biochemical activity. Site directed mutagenesis of an active site residue predicted to have changed the enzymes specificity from a hydroxylase to a halogenase also confirmed this prediction. Based on phylogenetic reconstructions the authors further show that the enzyme evolved from a plant enzyme (i.e. independently of its bacterial counterparts) rather than it was gained through lateral gene transfer.

Overall these findings are significant for multiple reasons. First, having available a halogenase of plant origin opens new possibilities in plant synthetic biology and chemical engineering. Second, this is yet another example of convergent evolution in plant secondary metabolism and further cements the notion that few evolutionary changes may be sufficient to 'create' entirely new biochemical pathways.

I have few suggestions that could improve the manuscript. The phylogeny is somewhat minimalistic, but serves the point that the enzyme evolved fairly recently through gene duplication rather than through lateral gene transfer. Expanding the species range within the Menispermaceae (and expanding it into other Ranunculales) would have more convincingly showed when the

duplication happened within this family / order.

The targeted mutagenesis (G226D) experiments clearly showed that G226 is necessary for halogenase activity, but it does not show that the D → G mutation was sufficient to gain, at least some, halogenase activity. Introducing this mutation into the sister FLS-like sequence (or better into the immediate pre-duplication ancestor (which could have been reconstructed from the [expanded] phylogeny) would be an interesting experiment.

Overall, the manuscript is written very well and I have essentially no editorial suggestions. Only the sentence "As predicted by this mechanism, mutating Ala226 back to Asp readily converts DAH to a hydroxylase" appears unclear. What does this refer to?

Point-by-point responses to reviewers' comments are in blue.

Reviewer #1 (Remarks to the Author):

*The authors describe the discovery of a 2-oxoglutarate/Fe(II)-dependent halogenase of plant origin. The identified enzyme, named dechloroacutumine halogenase (DAH), catalyzes the terminal chlorination step in the biosynthesis of the chloroalkaloid (-)-acutumine in plants of the Menispermaceae family. The corresponding genes encoding DAH in *Sinomenium acutum* and *Menispermum canadense* were identified by comparative transcriptomic analyses versus the non-producer *Stephania japonica*, by tissue specific analyses (root vs. leaf tissues) of all three plant species and by probing for iron(II)- and 2-oxoglutarate-dependent dioxygenases. Ordering synthetic genes for the identified enzymes, the authors were able to express functional DAH enzymes from *S. acutum* and *M. canadense* in *E. coli* and could show halogenation activity *in vitro*. Furthermore, the authors highlighted anion promiscuity of the enzymes SaDAH and gave evidence that alternative catalytic activities can be accessed by SaDAH and its variants (azidation and hydroxylation).*

The research is well described and clearly presented and the manuscript's findings are a valuable contribution to the field as the discovery of a freestanding 2-oxoglutarate/Fe(II)-dependent halogenases of plant origin are expanding the (so far very small) biocatalytic toolbox of these type of halogenases.

However, prior to publication in Nature Communications, there are a number of concerns which need to be addressed:

Major points

(1) The authors show that the newly discovered enzymes exhibit a certain anion promiscuity such as for NaN₃. To increase the depth of the study, it would be interesting to also learn more about the enzymes' substrate promiscuity, for example with respect to related alkaloids such as hasubanonine.

We thank the reviewer for this suggestion. In the first submission of our manuscript, we tested DAH's substrate promiscuity with a structural analog of dechloroacutumine, sinomenine, which DAH did not have any activity on. To increase the depth of this study, we examined substrate promiscuity to other alkaloids—codeine, berberine, boldine, scoulerine and vinpocetine. We show the subsequent LC-HRAM-MS data highlighting the DAH *in vitro* activities in Supplementary Figure 8. We hoped to include hasubanan-type alkaloids in this list; however, they were either not commercially available or extremely difficult to obtain from chemical synthesis. Overall, we conclude from our *in vitro* assays that DAH has limited substrate promiscuity and harbors specific activity to its native substrate dechloroacutumine.

The following sentence has been added in our main text to reflect this examination:

“To examine the potential substrate promiscuity of SaDAH, we tested its halogenase activity against a panel of additional alkaloids including vinpocetine, codeine, scoulerine, berberine, and boldine. No halogenated products could be detected in these assays, suggesting that SaDAH is likely a specific halogenase towards its native substrate (–)-dechloroacutumine (**Supplementary Fig. 8**).”

And the addition of Supplementary Figure 8:

“**Supplementary Figure 8 | TIC chromatograms of SaDAH substrate promiscuity tests.** SaDAH-WT chlorinase activity was tested on structurally similar alkaloids shown in (a). No chlorinated product was identified in the +2OG assay (blue) compared to the -2OG negative control assay (black) at the presence of 200 μ M of (b) vinpocetine, (c) codeine, (d) scoulerine, (e) berberine, and (f) boldine. The displayed TIC mass window is 300-400 m/z .”

(2) *The depicted LCMS analyses are extracted ion chromatograms. It would be important to also show a TIC of the in vitro chlorination to rule out other major side products.*

We added the TIC of the *in vitro* chlorination of dechloroacutumine by DAH as a panel in Supplementary Figure 5b. Since no other distinct peaks were observed in the TIC, we conclude that there were no other major side products from the chlorination assays. This strengthens our characterization of DAH, as it confirms its specific chlorination activity on dechloroacutumine.

The following sentence has been added to the caption of Supplementary Figure 5:
“(b) TIC chromatogram of the SaDAH *in vitro* enzyme assay. The peaks corresponding to (–)-acutumine and (–)-dechloroacutumine are indicated as (1) and (5), respectively. There are no side products observed in this assay.”

(3) *Supplementary information / Supplementary Figure 11: Dechlorination of (-)-acutumine: Recorded spectrometric data should be shown, purity of obtained product should be described.*

Due to limited availability of the reaction substrate to yield an adequate amount of (–)-dechloroacutumine for NMR structural characterization, we have instead included a MS/MS spectral matching of the (–)-dechloroacutumine to the (–)-dechloroacutumine detected in the synthetic sample that Herzon Lab (Yale) shared with us. The fragmentation similarity score between the two spectra is 0.7853 and it supports the production of (–)-dechloroacutumine. Furthermore, we show XICs of (–)-dechloroacutumine (364.17 m/z) for our reaction product and Herzon Lab’s synthetic sample, which indicate a similar retention time of 3.47 min. We also show a TIC chromatogram (scanning mass range 100 - 600 m/z) of the purified product in Supplementary Figure 14 that qualitatively describes the purity of (–)-dechloroacutumine.

We have added the following plots in Supplementary Figure 14, along with the caption as follows:

(b) The MS/MS fragmentation pattern of (5) is in agreement with the spectral values by King, *et al*⁶. The spectral cosine similarity score of 0.7853 was calculated using the OrgMassSpec v0.4-4 package in R with m/z tolerance at 0.005 and baseline threshold at

34%. (c) TIC of the purified (5) shows no significant side-products from the dechlorination reaction and purification process. (d) XICs of 5 (364.17 *m/z*) for product from the dechlorination reaction (black) and the synthetic mixture from King, *et al.* (red).

(4) *The authors should give a more detailed description in the main text how the proposed (-)-acutumine biosynthetic pathway was elucidated (Figure S1). Did the differential expression analysis in fact only yield one possible enzyme sequence candidate for each biosynthetic step as suggested in Figure 1c and S2?*

We apologize for any confusion regarding the proposed biosynthetic pathway. Supplementary Figure 1 is a proposal of the (-)-acutumine biosynthetic pathway that was determined based on the findings from a previous D₂O labeling study (Nett *et al.*, *AiChE* 2018). We propose this pathway because we believe that (-)-acutumine is most likely derived from the BIA pathway at certain points and we searched for known BIA enzymes that use reticuline as a precursor in the pathway. While the whole pathway is interesting and involves novel chemistry, it is beyond the scope of this manuscript.

To clarify this point, a more detailed description of how we proposed the (-)-acutumine biosynthetic pathway has been added. The changes are:

In the main text:

“Using D₂O-labeling technique, a previous study explored the plausible (-)-acutumine biosynthetic pathway by tracing deuterium-labeled benzylisoquinoline alkaloids (BIAs) in *M. canadense*¹⁹. Based on this proposed pathway and prior knowledge about BIA metabolism in other plants, we examined functionally annotated transcripts that are highly and differentially expressed in the root versus the leaf or stem tissue of *M. canadense*, from which numerous candidate genes for the biosynthesis of (-)-acutumine and other related BIAs were identified.”

In Supplementary Figure 1 caption:

“Enzymes with high expression in *M. canadense* root tissue and annotation to BIA biosynthesis were chosen as the representative candidates, but remain uncharacterized.”

(5) *Fe(II)-dependent halogenases have been previously described to exhibit hydroxylation activity (as also observed by the authors for the DAH enzymes). Importantly however, regioselectivity of CH-activation may differ depending on the reaction chemistry (e.g. hydroxylation vs. halogenation) as selectivity is determined by the orientation of the substrate with respect to the HO-Fe-Cl (Mathews *et al.*, *PNAS* 2009, 106, 17723-17728; Hayashi *et al.*, *ANIE* 2019, 58, 18535-18539). Whereas the authors do have an acutumine standard to which they compare the obtained MS and MS2 data, such a standard is not described for the azide and hydroxylated (side) products. Nevertheless, the authors postulate a structure for the obtained hydroxy and azido derivatives of acutumine (Figure 2a, Figure S9). Contrary to the*

author's argument in the Figure legend S9, in light of the findings of above mentioned papers structural data is required to verify regio-and stereospecificity of the proposed products.

While we agree with the reviewer and acknowledge the absence of structural data, we feel confident in the putative regio-assignment of the -OH and -N₃ products and have already stated our uncertainty in their stereo-chemistry. The above mentioned references look at reactivity with unnatural substrates where the enzyme is shown to target multiple C-H regio-centers. However, when -OH vs. -X transfer has been explored with native enzymes that strongly favor a single C-H bond, the -OH and -X installation has been observed to remain at that specific regio-center (Mathews et al, PNAS 2009; Mitchell et al, Nat. Chem. Bio. 2016; Mitchell et al, Biochem 2017). Due to limited availability of the reaction substrate for this study, producing enough of the -OH and -N₃ for NMR structural characterization will be costly and difficult. We recognize the importance of claiming certainty for publication and have changed the descriptive text to reflect this. The changes are as follows:

In main text:

“The regio- and stereo-specificity of the -OH and -N₃ moieties is not directly determined, but can be reasonably postulated to be at the C11 position of (–)-dechloroacutumine with the same stereochemistry as the -Cl group in (–)-acutumine (**Fig. 2a**, **Supplementary Fig. 12**). The fact that only a single -Cl product is observed in SaDAH-WT chlorination assay indicates that the enzyme is not capable of abstracting an alternate C-H bond (**Fig. 2b**, **Supplementary Fig. 5**). This postulation is consistent with the nature of substrate binding and C-H bond abstraction previously reported in other bacterial 2ODHs²⁰.”

In Figure 2 caption:

“The putative regiochemistry of the -OH and -N₃ in (**7**) and (**8**) are highlighted, while the stereochemistry of the products remains unknown.”

In Supplementary Figure 12 caption:

“Moreover, the *m/z* transitions from 323.11->305.10 in **7** and 320.11->305.10 in **8** correspond to the loss of -OH and N•, respectively. This observation is consistent with the *m/z* transition from 341.08->305.10 in **1** that has been previously reported in a proposed loss of -Cl in MS² analysis of **1**⁵ and leads us to support the regiochemical assignments of **7** and **8**.”

(6) What is the total turnover number of the DAH enzymes?

The total turnover numbers of the DAH enzymes are difficult to obtain as the prolonged activity assay will require a substantial amount of (–)-dechloroacutumine, but it is extremely difficult to obtain such quantity to set up this experiment. There is no visible decline of activity in a reasonable amount of time (~12 hrs) with the amount of substrate (100 μM) we used in our

enzyme assays. Instead of the total turnover number, we report the kinetic parameters to provide insights into the DAH enzyme in our earlier submission:

“SaDAH exhibits a K_M of $18.35 \pm 7.13 \mu\text{M}$ against (–)-dechloroacutumine as the substrate, and an apparent k_{cat} of $63.44 \pm 8.75 \text{ min}^{-1}$ (**Supplementary Fig. 8**), which are comparable to the kinetic constants measured for previously characterized bacterial 2ODHs²⁴ and plant 2ODDs²⁷.”

(7) *Is there any evidence for the proposed gene duplication event?*

We appreciate the reviewer’s attention to the proposal of gene duplication. To further test the gene duplication hypothesis, we expanded our phylogenetic analysis of DAH with additional 2ODD homologs found in closely-related species to *Menispermaceae* (Supplementary Fig. 11). This new analysis shows strong evidence supporting a recent gene duplication event from a flavonol synthase-like progenitor that gave rise to DAH after *Menispermaceae* diverged from the other plants under *Ranunculales*.

The following text has been added in the main text:

“Moreover, an extended phylogenetic analysis including 2ODD homologs identified from the available transcriptomes of *Ranunculales* plants further supports that DAH recently evolved from an FLS-like progenitor through a gene duplication event that occurred in *Menispermaceae* plants after they diverged from the rest of *Ranunculales* plants (**Supplementary Fig. 11**).”

Supplementary Figure 11 has been added, along with the following text:

“Supplementary Figure 11 | Extended phylogenetic tree analysis of DAH with other select plant 2ODD sequences. The phylogenetic tree of DAHs together with select 2ODD-family proteins is inferred using Maximum-likelihood method. Bootstrap statistics (200 replicates) greater than 70% are indicated at the tree nodes. The scale measures evolutionary distance in substitutions per amino acid. Asterisks denote the two orthologous DAHs identified in this study, along with the location of the proposed gene duplication event in red. The green branches indicate FLS-like sequences from *Ranunculales* plants and *A. thaliana*. The multiple sequence alignment used for building the phylogenetic tree is in Supplementary File 2. Abbreviation: FLS, flavonol synthase.”

(8) *The authors conducted transcriptome analysis of all three plant species but only in case of M. canadense, this analysis was performed on different plant organs (root, leaf, stem). Does the plant S. acutum also show a difference in acutumine production in roots versus leaf?*

This is a good point raised by the reviewer. The *S. acutum* and *S. japonica* plants at the UConn Greenhouses were dormant at the time of tissue collection. Therefore, only root material was available. Past research has established (–)-acutumine accumulation in the roots of another *Menispermaceae* species (Babiker et al, Bioscience, Biotechnology, and Biochemistry 1999). While it is interesting to observe a difference in (–)-acutumine production in roots vs.

leaves in *S. acutum*, we believe that the findings would not significantly enrich the main findings of our study.

(9) The authors should state more clearly, why SaDAH was chosen as the representative DAH for further analysis.

SaDAH was chosen as the representative DAH for further analysis due to better accessibility, along with higher protein expression yield in our purification experiments. Additionally, *S. acutum* transcriptome was the first assembled transcriptome in this study. We show in Supplementary Figure 6 that McDAH has identical enzymatic activity to that of SaDAH. We also show that SaDAH and McDAH have 99.1% sequence similarities with 3 amino acid differences. Thus, there is no significant reason as to why SaDAH had to be the representative DAH other than convenience in our study. The main text was changed accordingly to state this more clearly:

“Recombinant McDAH exhibits identical activity to SaDAH (**Supplementary Fig. 7**). Since the recombinant protein yield was higher for SaDAH than McDAH, we chose SaDAH as the representative DAH for detailed characterizations hereafter.”

Other (minor) points:

(1) Page 2: “To identify the candidate enzyme responsible for the terminal chlorination, we probed several known classes of halogenases, including heme haloperoxidases¹⁶, vanadium-dependent haloperoxidases²⁰, flavin-dependent halogenases¹⁸, S-adenosyl-L-Met-dependent halogenases²¹, methyl halide transferases¹⁶, and iron(II)- and 2-oxoglutarate-dependent halogenases (2ODHs)²².” This sentence is confusing. The authors should describe shortly in which way they “probed” the obtained transcripts for all of the enumerated enzyme classes.

We have changed the text to reflect more accurately our thought process and methods: “We postulated that the candidate enzyme responsible for the terminal chlorination was catalyzed by an 2-oxoglutarate-dependent halogenase (2ODH) because it is the only characterized class of halogenases capable of targeting an unactivated sp^3 -hybridized carbon centers for single-step halogenation²⁰. We therefore sought highly expressed candidate genes in our transcriptomes that are annotated as 2-oxoglutarate-dependent dioxygenases (2ODDs), and manually searched for the potential presence of active-site sequence variation unique to the 2ODH subfamily.”

(2) *Supplementary information: Protein expression and purification*

The authors should include a reference SDS-Page gel image to show purity of the wildtype DAH halogenases and the constructed variants.

We agree that a reference SDS-PAGE gel image will ensure purity of WT DAH and G226D DAH mutant. A gel image of the protein expression and purification of SaDAH has been added as Supplementary Figure 4 with the following caption:

“Supplementary Figure 4 | Protein expression and purity of recombinant SaDAH, McDAH, and SaDAH-G226D variant. The lanes in the gel correspond to the following: (1) SaDAH with 8xHis-tag, (2) SaDAH without tag, (3) McDAH with 8x His-tag, (4) McDAH without tag, (5) SaDAH-G226D with 8xHis-tag, (6) SaDAH-G226D without tag. The asterisk indicates the lane for BlueStain™ protein ladder (Goldbio).”

(3) Supplementary information: SaDAH-WT kinetic assay

How was the internal standard curve (obtained by quenching of enzymatic reaction after 3h hours) verified? Did the authors compare with integrated peak areas of product solutions with known concentrations to ensure that conversion of substrate to product was indeed quantitative?

The internal standard curve was generated by assaying the substrate solutions with known concentrations until full conversion (3 hours). In preliminary experiments, the complete conversion of the substrate at these chosen concentrations were observed within 2 hours, and no additional changes were observed with prolonged incubation at the end of 3 hours. As a practical consideration for enzyme kinetic assays, a product standard curve (especially for lower concentration range) could be generated by using the identical assay setup with substrate of known concentrations to be completely converted to product. In this case, we have verified in preliminary experiments that the conversion of the substrate is quantitative and complete by the end of 3 h incubation. Using serially diluted stock solutions of the product to make internal standard curve is a viable alternative approach, but in practice is less accurate compared to our approach for controlling errors caused by differential compound purities, initial weighing, and serial dilution for the substrate and product compounds.

(4) Supplementary Figure 2 depicts a differential expression analysis of M-canadense root versus stem transcriptomes. What is the difference / interpretation of this data compared to the data presented in the main article (root versus leaf)?

There is no qualitative difference in the interpretation of the tissue-specific expression analysis of *M. canadense* between root vs. stem and root vs. leaf. Since (-)-acutumine is significantly more enriched in the root than stem and root of *M. canadense*, Supplementary Figure 2 provides another evidence that the differential expression analysis in tissue-specific transcriptomes can enable gene discovery associated with its biosynthetic pathway.

(5) Page 3: The name of the species should be written out with first mentioning (Menispermum canadense)

Thank you for catching this error. We have made the correction in the main text.

(6) The following sentence needs to be clarified, as in DAH enzymes alanine corresponds to the wildtype.

“As predicted by this mechanism, mutating Ala226 back to Asp readily converts DAH to a hydroxylase”

We apologize for this error and lack of clarity. It should have been Gly226 instead of Ala226. The text has been improved as follows:

“As predicted by this mechanism, the mutation of Gly226 of DAH to an Asp residue restores the canonical 2ODD iron-binding facial triad, thus preventing halide coordination and revert the reaction outcome to hydroxylation²⁰.”

Reviewer #2 (Remarks to the Author):

This is an excellent paper describing well executed research that led to the isolation and partial characterization of a truly novel 2-oxoglutarate- and iron-dependent dioxygenase that catalyzes the chlorination of a alkaloid substrate as the final catalytic step in the formation of the unique chloroalkaloid (-)-acutumine in plants of the Menispermaceae family. The strategy to identify gene candidates and the biochemical characterization of the isolated enzyme are clever and scientifically sound, respectively. The plant biology aspects of the work are generally limited to a comparative transcriptional analysis of the newly isolated halogenase, which displays a similar root- versus stem-enhanced expression compared with uncharacterized orthologs putatively encoding other alkaloid biosynthetic enzymes. Although most of these orthologs are likely to encode the proposed enzymes, this caveat should be better defined in the manuscript. It would clearly be nice to include a gene knockdown-type experiment to further confirm the physiological relevance of the enzyme, but this is clearly and generally difficult in non-model plant systems.

We thank the reviewer for this suggestion. However, the *S. acutum* sampled for this study were properties of the EEB Biodiversity Education & Research Greenhouses at University of Connecticut. We obtained permission for one-time sampling of the live tissues for genomic and metabolomic analyses. *M. canadense* plants were obtained from the Toadshade Wildflower Farm with very limited supply. Due to the limited access to *Menispermaceae* plants, it is currently not feasible for us to conduct well-controlled gene knockdown-type experiments on these plants. Moreover, as the reviewer pointed out, gene knockdown-type experiments in non-model plant systems are generally difficult to achieve in a time-sensitive manner. In the long run, we do seek to develop in-house rearing of these plants and/or tissue culture systems to enable knockdown-type experiments for investigating the whole acutumine biosynthetic pathway. Thus, we have added the following text in our discussion:

“Future experiments to knockdown expression of DAH and other candidate BIA metabolic enzyme-encoding genes in Menispermaceae plants not only will help elucidate the (-)-acutumine biosynthetic pathway, but also will shed light on the functional significance of (-)-acutumine accumulation to its native plant hosts.”

Also to better state other alkaloid biosynthetic enzymes in the proposed biosynthesis of (-)-acutumine, we have added the following text in Supplementary Figure 1 caption:

“Enzymes with high expression in *M. canadense* root tissue and annotation to BIA biosynthesis were chosen as representative candidates, but remain functionally uncharacterized.”

From a biochemical perspective, it would be interesting to include empirical data showing the substrate specificity, or lack thereof, of the novel halogenase. Other basic aspects related to the characterization of the enzyme are strong, and the discussion is pertinent and informative.

To survey the substrate specificity/promiscuity of DAH, in addition to sinomenine, we have now tested DAH's halogenase activity against a panel of alkaloids, including codeine, berberine, boldine, scoulerine and vincetamine. We have added Supplementary Figure 8 to show that DAH activity is specific to its native substrate, (-)-dechloroacutumine, whereas it shows no detectable halogenase activity towards the other substrates tested.

The following sentence has been added in our main text to reflect this additional result: "The substrate promiscuity of SaDAH was further examined by testing its activity against a panel of additional alkaloids including vincetamine, codeine, scoulerine, berberine, and boldine. No halogenated products could be detected in these assays, suggesting that SaDAH is likely a specific halogenase towards its native substrate (-)-dechloroacutumine (**Supplementary Fig. 8**)."

And the addition of Supplementary Figure 8:

"Supplementary Figure 8 | TIC chromatograms of SaDAH substrate promiscuity tests. SaDAH chlorinase activity was tested on structurally similar alkaloids shown in (a). No chlorinated product was identified in the +2OG assay (blue) compared to the -2OG negative control assay (black) at the presence of 200 μ M of (b) vincetamine, (c) codeine, (d) scoulerine, (e) berberine, and (f) boldine. The displayed TIC mass window is 300-400 m/z."

Reviewer #3 (Remarks to the Author):

The authors reported the discovery of first enzyme of plant origin involved in halogenation, which is rare and would be an important biotechnological tool for expanding chemodiversity of plant natural products. I suppose that the impact and novelty of the finding reach the standard of the journal and are presented well generally. Only concern of mine is the amount of data, which seem fair for brief, but not for full, article. I hope that additional experiments suggested below would help to improve the study in that direction.

We agree with the reviewer's assessment of the nature of this manuscript and the discovery described within, as we also intended this paper to be published as a communication. The "article" assignment was given by default by the submission system. As Nature Communications welcomes papers with all formats, we will work with the editors to format this manuscript to the most appropriate format suited for the journal.

1, They showed that SaDAH-G226H catalyzes the hydroxylation rather than halogenation. Is it plausible to confer the halogenation activity to related ODDs with a substitution HxG? This issue is of great interest, considering the generation of novel halogenation catalyts.

We thank the reviewer for this suggestion. Indeed, we are currently pursuing this interest in a wide range of plant ODDs, but believe the results of this pursuit are more suitable for a follow-up paper. This idea was originally piqued upon the identification of the the first 2ODD-family halogenase, but our current set of obtained results suggested that the simple HxG mutation is not sufficient to fully confer halogenation in several plant 2ODDs tested. Thus far, there has only been a single successful example of converting a 2ODD hydroxylase into a halogenase (Mitchell et al, Biochem 2017) and at very poor efficiency. We hope the identification of DAH in this study and our future work along this line will ultimately help efficient engineering of 2ODD-family halogenases.

2, In planta experimental evidences are lacking at all. Better to add the experiments technically possible to make the study more sound and lengthy enough, I believe.

We agree that *in planta* experiments such as gene-knockdown experiment would test the necessity of *DAH* in acutumine production in acutumine-producing *Menispermaceae* plants. However, as in response to one of the Reviewer #2's comments, such experiment is not feasible at this moment (see above response to Reviewer #2). In our paper, we therefore avoid making any claim about the necessity and sufficiency of *DAH* in acutumine biosynthesis in native plant hosts. In future research to elucidate the whole acutumine biosynthetic pathway, we seek to establish in-house cultivation of model *Menispermaceae* plants and explore methods for gene-knockdown and gene-in experiment in these non-model plants.

3, It would be better to deliver a reason (more enriched (-)-acutumine accumulation in the roots) before mentioning the selection of candidate transcripts based on their high expression in the roots.

We thank the reviewer for bringing this to our attention. The following changes have been made in our main text to reflect this:

“Furthermore, we found that (–)-acutumine is more enriched in roots compared to stems and leafs in *M. canadense*.”

Reviewer #4 (Remarks to the Author):

*The work by Kim et al targets a rare yet important plant secondary metabolic pathway. Halogenated organic compound are frequent in pharmaceuticals and many other synthetic organic compounds, but are rarely found in plants. The authors targeted one these rare plant groups and used a simple yet elegant approach to identify candidate genes. They combined a simple comparative root versus leaf transcriptomics approach with candidate gene family selection and biochemical rational to identify a single candidate gene to catalyze the final chlorination reaction to produce the halogenated compound, called acutumine. Heterologous expression of the candidate protein in *E. coli* validated the predicted biochemical activity. Site directed mutagenesis of an active site residue predicted to have changed the enzymes*

specificity from a hydroxylase to a halogenase also confirmed this prediction. Based on phylogenetic reconstructions the authors further show that the enzyme evolved from a plant enzyme (i.e. independently of its bacterial counterparts) rather than it was gained through lateral gene transfer.

Overall these findings are significant for multiple reasons. First, having available a halogenase of plant origin opens new possibilities in plant synthetic biology and chemical engineering. Second, this is yet another example of convergent evolution in plant secondary metabolism and further cements the notion that few evolutionary changes may be sufficient to 'create' entirely new biochemical pathways.

I have few suggestions that could improve the manuscript. The phylogeny is somewhat minimalistic, but serves the point that the enzyme evolved fairly recently through gene duplication rather than through lateral gene transfer. Expanding the species range within the Menispermaceae (and expanding it into other Ranunculales) would have more convincingly showed when the duplication happened within this family / order.

This is an excellent suggestion made by the reviewer. We have expanded our phylogenetic analysis to include 2ODD proteins from sequenced genomes of *Physcomitrella patens*, *Selaginella moellendorffii* and *Arabidopsis thaliana*, as well as available transcriptomes of *Ranunculales* family in the 1kp database. This updated analysis, shown as Supplementary Figure 11, further supports that DAH is derived from a recent gene duplication event from an flavonol synthase-like progenitor after the divergence of the *Ranunculales* family of plants.

The following text has been added in the main text:

“Moreover, an extended phylogenetic analysis including 2ODD homologs identified from the available transcriptomes of Ranunculales plants further supports that DAH recently evolved from an FLS-like progenitor through a gene duplication event that occurred in Menispermaceae plants after they diverged from the rest of Ranunculales plants (**Supplementary Fig. 11**).”

Supplementary Figure 11 has been added, along with the following text:

“Supplementary Figure 11 | Extended phylogenetic tree analysis of DAH with other select plant 2ODD sequences.

The phylogenetic tree of DAHs together with select 2ODD-family proteins is inferred using Maximum-likelihood method. Bootstrap statistics (200 replicates) greater than 70% are indicated at the tree nodes. The scale measures evolutionary distance in substitutions per amino acid. Asterisks denote the two orthologous DAHs identified in this study, along with the location of the proposed gene duplication event in red. The green branches indicate FLS-like sequences from Ranunculales plants and *A. thaliana*. The multiple sequence alignment used for building the phylogenetic tree is in Supplementary File 2. Abbreviation: FLS, flavonol synthase.”

The targeted mutagenesis (G226D) experiments clearly showed that G226 is necessary for halogenase activity, but it does not show that the D -> G mutation was sufficient to gain, at least some, halogenase activity. Introducing this mutation into the sister FLS-like sequence (or better into the immediate pre-duplication ancestor (which could have been reconstructed from the [expanded] phylogeny) would be an interesting experiment.

We appreciate the reviewer's suggestion for this experiment. As in response to a similar comment from Reviewer #3's comment, we are actively pursuing this idea that will constitute a separate manuscript (see above response to Reviewer #3).

Overall, the manuscript is written very well and I have essentially no editorial suggestions. Only the sentence "As predicted by this mechanism, mutating Ala226 back to Asp readily converts DAH to a hydroxylase" appears unclear. What does this refer to?

We thank the reviewer's attention to this particular error. There is a typo— it should be Gly226 instead of Ala226. This refers to the SaDAH-G226D variant that was made to showcase its hydroxylation activity. This has been corrected in the main text.

REVIEWERS' COMMENTS:

Reviewer #1 (Remarks to the Author):

The authors addressed all my concerns in their revised manuscript. I recommend publication in Nature communications.

Reviewer #3 (Remarks to the Author):

I understand the technical difficulty associated with non-model plant materials. I suppose that authors responded to all comments sincerely and their excuses are largely understandable. One concern is the type of the publication. I would like to leave this point to editor's dealing.

Reviewer #4 (Remarks to the Author):

The authors addressed my points to my satisfaction with exception of the timing of the gene duplication that lead to DAH. The extended phylogeny nicely clarifies in more detail when the gene duplication took place. However, I disagree that the duplication "occured in Menispermaceae plants after they diverged from the rest of Ranunculales plants". The DAH sequences group at the base of the Ranunculales FLS-like clade (without bootstrap support...), which presumably includes Menispermaceae and other families. This would suggest that the duplication happened early after separation of the Ranunculales from other eudicot lineages, but before Menispermaceae separated from other lineages (families) within the Ranunculales. If the duplication happened within the Menispermaceae, the DAH sequences should have been part of a (or the) Menispermaceae clade within the Ranunculales. It may be helpful to annotate clades within the Ranunculales with taxonomic family information. It appears Supplemental Data Table 2 detailing the sequence information used for the extended phylogeny is missing.

Reviewer #1 (Remarks to the Author):

The authors addressed all my concerns in their revised manuscript. I recommend publication in Nature communications.

Thank you.

Reviewer #3 (Remarks to the Author):

I understand the technical difficulty associated with non-model plant materials. I suppose that authors responded to all comments sincerely and their excuses are largely understandable. One concern is the type of the publication. I would like to leave this point to editor's dealing.

Thank you.

Reviewer #4 (Remarks to the Author):

The authors addressed my points to my satisfaction with exception of the timing of the gene duplication that lead to DAH. The extended phylogeny nicely clarifies in more detail when the gene duplication took place. However, I disagree that the duplication “occured in Menispermaceae plants after they diverged from the rest of Ranunculales plants”. The DAH sequences group at the base of the Ranunculales FLS-like clade (without bootstrap support...), which presumably includes Menispermaceae and other families. This would suggest that the duplication happened early after separation of the Ranunculales from other eudicot lineages, but before Menispermaceae separated from other lineages (families) within the Ranunculales. If the duplication happened within the Menispermaceae, the ADH sequences should have been part of a (or the) Menispermaceae clade within the Ranunculales. It may be helpful to annotate clades within the Ranunculales with taxonomic family information. It appears Supplemental Data Table 2 detailing the sequence information used for the extended phylogeny is missing.

We thank the reviewer for catching this and agree with the reviewer that we cannot confirm the exact timing of this gene duplication event without genome sequencing results of

these plants. However, we found no clear orthologous sequences of DAH with the characteristic active site Asp->Gly mutation across Ranunculales plant transcriptomes. Thus, we claim that this gene has uniquely been evolved in (–)-acutumine producing plants within the Menispermaceae family. We have improved upon this expanded phylogenetic tree with taxonomic family information. We made the following changes in the main text and Supplementary Figure 11 to reflect this and also attached the alignment file as Supplementary Dataset 2.

“Moreover, an extended phylogenetic analysis including 2ODD homologs identified from the available transcriptomes of Ranunculales plants further supports that DAH recently evolved from an FLS-like progenitor through a gene duplication event that occurred during the radiation of Ranunculales, contributing to the occurrence of (–)-acutumine biosynthesis in Menispermaceae (**Supplementary Fig. 11, Supplementary Data 2**).”